# Modeling the Capacitated Multi-Level Lot-Sizing Problem under Time-Varying Environments and a Fix-and-Optimize Solution Approach

**DOI:** 10.3390/e21040377

**Published:** 2019-04-07

**Authors:** Meng You, Yiyong Xiao, Siyue Zhang, Shenghan Zhou, Pei Yang, Xing Pan

**Affiliations:** School of Reliability and System Engineering, Beihang University, Beijing 100191, China

**Keywords:** capacitated lot-sizing problem, time-varying environment, mixed-integer linear programming, optimization, entropy

## Abstract

In this study, we investigated the time-varying capacitated lot-sizing problem under a fast-changing production environment, where production factors such as the setup costs, inventory-holding costs, production capacities, or even material prices may be subject to continuous changes during the entire planning horizon. Traditional lot-sizing theorems and algorithms, which often assume a constant production environment, are no longer fit for this situation. We analyzed the time-varying environment of today’s agile enterprises and modeled the time-varying setup costs and the time-varying production capacities. Based on these, we presented two mixed-integer linear programming models for the time-varying capacitated single-level lot-sizing problem and the time-varying capacitated multi-level lot-sizing problem, respectively, with considerations on the impact of time-varying environments and dynamic capacity constraints. New properties of these models were analyzed on the solution’s feasibility and optimality. The solution quality was evaluated in terms of the entropy which indicated that the optimized production system had a lower value than that of the unoptimized one. A number of computational experiments were conducted on well-known benchmark problem instances using the AMPL/CPLEX to verify the proposed models and to test the computational effectiveness and efficiency, which showed that the new models are applicable to the time-varying environment. Two of the benchmark problems were updated with new best-known solutions in the experiments.

## 1. Introduction

The multi-level lot-sizing (MLLS) problem [1,2] plays an important role in the efficient operation of a modern manufacturing and assembly system. It involves determining the optimal production quantities and periods for a production system in order to balance the trade-off cost between the production setup and inventory-holding. The MLLS is also a key part of many planning systems of a manufacturing firm, including material request planning (MRP), manufacturing execution system (MES), capacity planning system (CPS), and the inventory planning system, and has great influence on the economic benefit of a manufacturing enterprise. Nevertheless, the MLLS problem is extremely difficult solve with optimality, as modern products may have multi-level structures with complex interdependencies. Arkin et al. [3] proved that the MLLS problem is strongly NP-hard. The optimal algorithms existing in the literature can only solve small-sized problems, such as the dynamic programming formulations proposed by Zangwill [4], the constructive method for assembly structure proposed by Crowston and Wagner [1], and the branch-and-bound-based algorithms [5,6]. 

The lot-sizing problem was originally raised by the cost reduction requirement of inventory management [7]. Since Harris [8] published the famous economic order quantity (EOQ) formula, which can be viewed as the earliest form of the single-point lot-sizing problem, the trade-off optimizations between the one-time ordering/setup cost and the continuously occurring inventory-hold cost has become a hot research topic in industries and academia for a century’s long time. In 1958, Wagner and Whitin [9] first introduced the single lot-sizing problem and proposed a dynamic programming algorithm, i.e., the well-known WW algorithm, as an exact solution approach for optimal solutions. Schussel [10] discussed the lot-sizing planning problem for linear production systems, which was the first study that considered the lot-sizing problem involving multi-level product structures. In the 1990s, the fast development of computer-integrated manufacturing system (CIMS) technologies promoted the popular use of the MRP system in manufacturing enterprises, and has attracted great academic attention for the optimization of the MLLS problem. According to the existence of capacity constraints, the models for MLLS can be divided into two categories: (1) the uncapacitated MLLS model and (2) the capacitated MLLS model. It is generally considered that the uncapacitated MLLS belongs to the category of classical MRP systems and the capacitated MLLS problem belongs to an MRP-II/MES system, and the latter was generally deemed as closer to actual production situations, with stronger practical significance.

The uncapacitated MLLS problem was formally formulated by Yelle [2], where the sum of production setup cost and holding cost occurring in multiple periods within a limited planning horizon was taken as the objective function to be minimized. In comparison to the single-level lot-sizing (SLLS) problem that involves optimizing only one item of its production quantities, the MLLS needs to determine the optimal production volumes and corresponding periods for multiple items (products and parts) that have complex interdependencies. The internal dependencies among items may cause the problem, a great computational complexity and may bring difficulty in finding the optimal solution. Steinberg and Napier [11] first proved that the uncapacitated MLLS is a HP-hard problem in an ordinary sense. There are also a number of works in the literature that studied the MLLS problem on products with only pure-assembly structures and utilized the internal triangular characteristics of the production setup decision variables to improve the solution quality [12,13,14,15]. Meta-heuristic algorithms such as the MAX-MIN ant colony optimization (ACO) systems [16], the particle swarm optimization (PSO) algorithm [17], the genetic algorithm [18,19], the soft optimization (SOA) approach based on segmentation [20,21], and the variable neighborhood search (VNS) algorithms [22,23] have been developed for solving the medium- and large-sized uncapacitated MLLS problems with general product structures. 

The capacitated MLLS problem involves additional constraints on the production capacities of the planning periods. That is, the total volume of products produced in one period is not allowed to exceed a predefined upper limit, or the excess parts are imposed with cost penalties [24,25,26,27]. Sahling et al. [28] presented a model considering the dual constraints of capacity and resources, in which they defined a “resource–product” relationship matrix to transfer the product demand into resource demand, and set an upper limit for each of the resources in each period. Almeder et al. [29] studied the capacitated MLLS problem with lead time consideration. They developed two models considering batch production and allowing lot-streaming. Wu et al. [30] developed an optimization framework for the capacitated MLLS problem with backlogging, where the customers’ demands were allowed to be postponed and satisfied in delayed periods with compensation penalties.

In addition to the production setup cost and inventory-holding cost, other types of cost, such as the production cost, switch-off cost, and line changeover cost, could also be considered in lot-sizing models in order to make them more adaptable to different production environments [31,32,33]. Most of the lot-sizing models in the literature have made an assumption that backloggings are not allowed, which means that the customers’ demands must be satisfied in their required periods without delays. However, in some cases, delayed deliveries must be allowed due to the impacts of various unexpected events such as machine breakdown, quality defects, and extreme weather. In these cases, the delayed deliveries were usually considered as weighted penalty costs added to the objective function [30]. Steigberg and Napier [11] transformed the MLLS problem in a general production system into a minimum cost flow problem, and obtained the optimal solution based on a network algorithm. However, the efficiency of their solution approach was relatively low. Zahorik and Thomas [34] analyzed the characteristics of the assembly system and then transformed the batch problem with three planning periods into a network problem. Blackburn and Millen [35] used the cost correction heuristic algorithm to solve the lot-sizing problem without capacity constraints. Homberger [19] designed a parallel genetic algorithm to solve the uncapacitated MLLS problem. Xie and Dong [36] studied a more general lot-sizing problem with a heuristic genetic algorithm as the solution approach. There are also other similar studies [37,38,39]. This MLLS optimization is not only treated as a production planning problem but also connected with the other subsystems of the manufacturing system, such as internal transport, material handling, and purchases [40,41]. In addition, Simulation methods are also frequently used for optimization problems in production environment [42,43,44].

However, all of the above works on lot-sizing problems had an assumption of a constant production environment where the unit costs of manufacturing resources were invariable, and the capacities were stable. The time-varying environment (TVE) is becoming common in many manufacturing industries because of the fast-changing market and individualized customer needs. In the modeling and biological literature, many types of time-varying environments are generally considered. The environmental types may include periodic, non-periodic, deterministic, stochastic, predictable environments, etc. In some more complex cases, exogenous variability may be affected by collective feedback from previous (delayed) dynamic activities. In TVE, production factors of the manufacturing system, such as the availability of manufacturing resources, material/labor costs/prices and production capacities are all potentially subject to continuous change. Due to the fierce competition environment, TVE exists widely in today’s manufacturing firms in many industries. Traditional lot-sizing theorems and algorithms are assumed for a constant production environment; thus, they are not applicable to TVE. There are only a few related works on lot-sizing problem with TVE that can be found in the literature. Martel and Gascon [45] first considered the difference of price and cost in different production periods. Haase and Kimms [46] considered that the production setup cost is dependent on the order of production steps. Dellaert and Jeunet [12,18], Haase and Kimms [46], and Raa and Aghezzaf [47] considered the fluctuation of external demand in different periods. Piperagkas et al. [48] and Tempelmeier and Hilger [49] studied the dynamic lot-sizing problem with stochastic demands over multiple planning periods. Chowdhury et al. [50] presented a new O(T) algorithm for the dynamic single-item lot-sizing problem. Related works in the literature on the lot-sizing problem under TVEs are still very limited.

Wiley et al. [51] first used the entropy theory to describe the structure of biological systems and defined the order degree of systematization. Li [52] discussed the proportional relationship between organization degree and the entropy of the system structure from the perspective of information quantity and gave the theoretical conditions for keeping the order of the system unchanged. Zhilin [53] established a structural entropy model for optimizing the structure of the system and calculating the order degree of the system structure by analyzing the influence of the system structure on the information flow in the system. All the studies agree with the fact that the more chaotic the production system is, the greater the entropy value of the system, while the entropy value of the optimized production system is generally smaller. Stephen et al. [54] studied entropy and provided contributions for modeling situated entropy in factories to improve the planning and control of such factories. Camelia [55] proposed a general model for the optimization of thermal machines with two heat reservoirs, which was applied to a Carnot-like refrigerator with non-linear heat transfer laws and internal and external irreversibility. Martina et al. [56] used a statistical-validation approach derived from maximum-entropy arguments to analyze the influence of technological innovations on industrial production. The above works applied entropy theory to factory manufacturing and are closely related to the subject of this paper. Except for its application in production systems, entropy theories have been widely used in supply chain systems, urban systems, groundwater systems, and other fields. Qiuyang et al. [57] analyzed the stability of equilibrium points by mathematical analysis and explored the influence of parameters on stability domain and entropy of a system. Yimin and Qiuxiang [58] analyzed and simulated the influences of decision variables and parameters on the stability and entropy of asymmetric dual-channel supply chain systems based on entropy theory. Yong et al. [59] analyzed the spatial structure of urban systems through entropy theory. Wei and Shanghai [60] described the evolution of a groundwater flow system with system information entropies. In addition, it is also very important to prepare optimized and robust production plans and schedules in order to reduce the necessity of re-scheduling during the manufacturing process. Related works can be found in Tolio et al. [61] and Sobaszek et al. [62]. The above works have made an important contribution to the application of entropy theory to production and manufacturing.

In this paper, we considered the time-varying setup cost and dynamic capacity constraints as new constraints based on the traditional capacitated MLLS framework. We present a new mixed-integer linear programming (MILP) model for the capacitated MLLS problem under a time-varying environment (CMLLS-TVE) and provide theoretical analysis on the model’s properties. Computational experiments were conducted on well-known benchmark problems and the model was solved in the AMPL/CPLEX programming environment. The experimental results showed that the AMPL/CPLEX solver could effectively solve the small-sized CMLLS-TVE problem instances with optimal solutions and could also solve a part of the medium-sized benchmark instances. However, for large-sized problem instances, we developed a fix-and-optimize heuristic algorithm based on partial optimizations to solve the CMLLS-TVE with near-optimal solutions within a controllable time limit.

The rest of the paper is organized as follows. In Section 2, we introduce the MILP model for the CMLLS-TVE, and the properties of the MILP model are discussed in Section 3. In Section 4, computational experiments are conducted to analyze the optimality of the solution with small-sized problems and the efficiency of the MILP model. In the same section, a fix-and-optimize heuristic approach is developed for medium- and large-sized problems. Finally, in Section 5, we summarize the paper and give further research directions.

## 2. Problem Description and Formulation

Traditional lot-sizing models assumed that the production environment was constant, and many production factors were considered to be consistent in multiple periods of the planning horizon, which are in fact not in accordance with today’s dynamic environment. For a manufacturing firm in TVE, a typical time-varying factor is the production setup cost that may not be a constant but a time-varying value depending on the length of time interval between the current setup and the previous one. This phenomenon can be explained that if the production interval is longer, then more costs (or time), such as cleaning, machine inspection, maintenance, and preparation, are needed for the production setup. The change of the setup costs can result from either outside or internal influences. The study considers all exogenous and endogenous environmental dynamics that may cause the cost variability.

### 2.1. Entropy in Production Systems

Entropy, one of the parameters in thermodynamics that characterize the state of matter, is represented by the symbol *S*. Its physical meaning is a measure of the chaos degree of the system. Entropy is ubiquitous in production systems. For example, the rationality of a factory layout, logistics transportation, processing procedure, and production task arrangement will affect the orderliness. Then, the entropy of the system is affected. In this paper, we considered the external demand and various factors in the production process to optimize the production schedule. The purpose of optimizing the production schedule is to make the production system more orderly. Correspondingly, the entropy of the system decreases. In this paper, the entropy value was adopted as a criterion for evaluating the effectiveness of optimization.

The entropy change of a system is formally expressed as $\Delta S=S_{B}-S_{A}$ [56,57], indicating the entropy change of the system from state *A* to state *B*. Since our models consider multi-level product structures and the objective function is to minimize the total cost, the system’s entropy value should be affected by the complexity of the product structure, the quantity of products produced, and the cost. Here, we propose a preliminary method for calculating the system entropy. Entropy of the system was assumed to be the product of these three (i.e., the complexity of product structure, the quantity of products produced, and the objective cost). This is a preliminary idea, which needs further improvement. Therefore, the formula of the entropy change rate of the system can be calculated by $\Delta R=(S_{B}-S_{A})/S_{A}$. It can be further expressed as $\Delta R=(PS_{B}PQ_{B}OC_{B}-PS_{A}PQ_{A}OC_{A})/PS_{A}PQ_{A}OC_{A}$, where the symbol **Δ***R* represents the entropy change rate, symbol *PS* represents the complexity of the product structure, symbol *PQ* represents the quantity of the products produced, and the symbol *OC* represents the objective cost. 

### 2.2. A Mathematical Programming Model for the CMLLS-TVE

We assumed a positive correlation between the setup cost and the production interval. For simplicity, the production setup cost was assumed to increase at a fixed rate for each unit of production interval. We introduced two new definitions for a product (say *i*), which were: (1) production interval, *L_it_*, of product *i* at period *t*, and (2) growth rate, *α_i_*, of setup cost of product *i*.

**Definition** **1.**
*Production interval L_it_: the number of periods between the period t and the last period arranged for production before t for product i, which can be mathematically expressed in Equation (1) as follows.*
(1)$$L_{it}=t-1-\max\{t^{\prime }\cdot y_{it}|1\leq t^{\prime }<t\},\;\forall t\neq 1,\;\forall i$$
*In Equation (1), y_it_ is a binary decision variable, indicating whether the product i is scheduled for production in the period t. If product i is arranged to be produced, then y_it_ = 1; otherwise, y_it_ = 0. This formula provides a method for calculating the production interval for each product i in each period. For the first period, where t = 1, we assume L_it_ = 0. For the second period, where t = 2, the production interval is also zero because the first period is always arranged with a production setup. This is in order to guarantee the solution’s feasibility. The value range of L_it_ for t ≥ 2 is [0, t − 1]. It should be noted that the production interval L_it_ is irrelative to the states of the production setup of production i in period t.*


**Definition** **2.***Growth rate of setup cost α_i_: the rate of the production setup cost increased with respect to one unit of production interval. This coefficient can be a fixed value determined in advance according to the actual production condition or be a time-varying parameter related to specified periods*. 

Based on the definitions above, the CMLLS-TVE problem can be described as follows. A set of product items, including end products, intermediate parts, and raw materials are going to be produced in a set of production periods over a planning horizon, in order to deliver the customers’ demands timely in their required periods. All items have known product structures which determine the intermediate dependencies and the leading time among them. The leading time is scheduled for the preparations by subsystems such as internal transports, material handling, purchases, etc., such that a full manufacturing chain can be integrated to be optimized as a whole in the CMLLS-TVE model. For each period, if a product is setup for production, then a setup cost occurs, and a batch quantity is calculated based on the available capacity in that period. All produced products will be stored as inventory in warehouses before being delivered to customers, which recurs an inventory-holding cost in each period. The production setup cost is considered a function of the production interval, proportional to the length of the production interval. The objective function of CMLLS-TVE is to arrange a set of optimized production setups in these periods in order to minimize the sum of the production setup cost and the inventory-holding cost over the entire planning horizon. 

Notations of parameters and decision variables used to describe the CMLLS-TVE problem are summarized in Table 1.

Problem CMLLS-TVE:

Minimize:(2)$$TC=\sum_{i=1}^{m}\sum_{t=1}^{n}\left(H_{it}I_{i,t}+s_{it}y_{it}+\alpha_{i}L_{it}y_{it}\right)$$

Subject to:(3)$$I_{it}=I_{i,t-1}+X_{it}-D_{it}\;\forall i,t$$
(4)$$L_{it}=t-1-\max\{t^{\prime }y_{it^{\prime }}|1\leq t^{\prime }<t\}\;\forall i,t>1$$
(5)$$L_{i1}=0\;\forall i$$
(6)$$D_{it}=d_{it}+\sum_{j\in \Gamma (i)}C_{ij}X_{j,t+l_{j}}\;\forall i|\Gamma_{i}\neq \emptyset ,\forall t$$
(7)$$y_{it}\geq y_{kt}\;\forall k\in \Gamma_{i}^{-1},\forall i,t$$
(8)$$X_{it}-M_{it}y_{it}\leq 0\;\forall i,t$$
(9)$$I_{it}\geq 0\;\forall i,t$$
(10)$$X_{it}\geq 0\;\forall i,t$$
(11)$$y_{it}\in \{0,1\}\;\forall i,t$$

In the above formulations, the objective function in Equation (2) represents the sum of the setup cost and the inventory-holding cost of all products/parts in all periods. Constraint (3) is the inventory-flow, representing the balance of inventory at the end of each period, which is subtracted by the demand and added by the production volume in the same period. Constraints (4) and (5) are designed to calculate the production interval *L_it_*, representing the number of periods between period *t* and the last period arranged for a production setup before *t*. The first period is initialized as zero. Constraint (6) indicates that the total demand of product *i* in period *t* is equal to the sum of the external demand in period *t* and the internal demand of products or components needed to assemble a higher level of products. Constraint (7) utilizes the rule introduced by Tang [13] for the case of pure assembly. That is, if the production of product *i* is not arranged to be produced in period *t*, then any predecessor product of product *i*, e.g., product *k*, should not be arranged to be produced in the same period. Constraint (8) guarantees that the production of product *i* takes place only in period *t*, which satisfies *y_it_* = 1, and restricts the production volume of products *i* in period *t* so that it does not exceed the maximum capacity limit *M_it_*. Constraint (9) indicates that backlogging is not allowed. Constraint (10) indicates that the production volume is non-negative. Constraint (11) defines *y_i,t_* as a binary decision variable, indicating if product *i* is arranged to be produced in period *t* (by *y_it_* = 1) or not (by *y_it_* = 0).

### 2.3. A Mixed-Integer Programming Model for the CMLLS-TVE

The mathematical programming model formulated in Equations (2)–(11) well illustrates the CMLLS-TVE problem; however, it is non-linear. In the following, we transform the non-linear model into a linear one so that optimal solutions can be obtained by using commercial MIP solvers such as Lingo and CPLEX. Note that the objective function in Equation (2) and Condition (4) in the CMLLS-TVE model are non-linear. To transform them into linear expressions, we defined a new variable to replace *L_it_*, as follows.

**Definition** **3.**
*Increment of setup cost u_it_: a binary decision variable indicating if product i has a unit of setup cost increased in period t (by u_it_ = 1) or not (by u_it_ = 0).*


The new variable *u_it_* was introduced to determine the impact of the production interval on the production setup cost in period *t*. It plays a similar role to *L_it_* in the optimization model but in a linear way. The MILP model of the CMLLS-TVE problem is presented as follows.

Problem CMLLS-TVE:

Minimize:(12)$$TC=\sum_{i=1}^{m}\sum_{t=1}^{n}\left(H_{it}I_{it}+s_{it}y_{it}+\alpha_{i}u_{it}\right)$$

Subject to:(13)$$I_{it}=I_{i,t-1}+X_{it}-D_{it}\;\forall i,t$$
(14)$$u_{it}\geq u_{i,t+1}+y_{i,t+1}-2y_{it}\;\forall \;t<n,i$$
(15)$$u_{in}=0\;\forall i$$
(16)$$D_{it}=d_{it}+\sum_{j\in \Gamma (i)}C_{ij}X_{j,t+l_{j}}\;\forall i|\Gamma_{i}\neq \emptyset ,\forall t$$
(17)$$X_{it}-M_{it}y_{it}\leq 0\;\forall i,t$$
(18)$$I_{it}\geq 0\;\forall i,t$$
(19)$$X_{it}\geq 0\;\forall i,t$$
(20)$$y_{it}\in \{0,1\}\;\forall i,t$$

Notations used in above model are defined in Table 1. The objective function in Equation (12) is to minimize the sum of the total production setup cost, the inventory-holding cost, and the extra cost recurred by the production interval. Constraint (14) and constraint (15) are designed to guarantee the linear relationship between variables *u_it_* and *y_it_.* Note that the variable *u_it_* does not have an increment of setup cost if there is no production setup after period *t*. So, *u_it_* = 1 always holds for the last period of each product *i*. Other constraints are kept from the model in Section 2.1. Note that the improved model for the CMLLS-TVE problem is linear, so it can be directly solved with optimality by using commercial MIP solvers such as Lingo and CPLEX.

## 3. Property Analysis of the CMLLS-TVE Model

We analyzed the relationships among the parameters and variables and propose several properties for the CMLLS-TVE problem, which can help to improve the computational efficiency of the heuristic solution approach by cutting off non-optimal solution spaces.

**Property** **1.**
*For an optimal solution of the CMLLS-TVE problem, the following expressions always holds:*
*(1)* 
$I_{i,t-1}\cdot y_{i,t}=0$
*(2)* 
$I_{i,t-1}\cdot X_{i,t}=0,\;\forall t>1,i$



**Proof.** This property can be proven by using a contradict method. Assuming that there is an optimal solution that has a certain period in which the inventory level is not zero and new production has been arranged at the same time, such that we have $I_{i,t-1}\cdot X_{i,t}>0$. We let the *I_i_*_,*t*−1_ part of the products to be produced in period *t*. Then, we have ${I^{\prime }}_{i,t-1}\cdot {X^{\prime }}_{i,t}=0$. Finally, we have a non-zero cost savings of $h_{t-1}\cdot I_{t-1}$. Thus, we have a new solution which is even better than the optimal solution, which is in contradiction with the assumption. Thus, the property holds. □

**Property** **2.***For an optimal solution of the CMLLS-TVE problem, the production volume in each period is calculated by*$X_{it}=y_{it}\sum_{j=t}^{k}D_{ij}$*, where*$k=\min\{t^{\prime }|t^{\prime }\geq t,y_{it^{\prime }}=1\},\forall i$.

**Proof.** Property 2 determines the minimum quantity of production volume in the period *t* when *y_it_* = 1, which covers exactly all customer demands behind period *t* and before the next production setup. This property can by be proven by using a contradict method. Assuming that there is an optimal solution in which more products are produced in a period than that determined by Property 2, then extra inventory-holding costs incur, and then this solution is not an optimal solution, which is contradictory to the assumption. □

**Property** **3.**
*For an optimal solution of the CMLLS-TVE problem, if the demand (D_i,t*_) for product i in period t^*^ is met by production setup in period t^**^, where X_i,t**_ = 1 and t^**^ < t^*^, then the demand in period t that t^*^ < t < t^**^ is also met by X_i,t**_.*


**Proof.** This is straightforward. As proved in the proof of the traditional model, if Property 1 is correct and the production interval does not change, then property 3 must be correct. □

**Property** **4.**
*For an optimal solution of the CMLLS-TVE problem, the expressions shown below always holds.*
(21)$$\sum_{t=1}^{n}\left(X_{i,t}\cdot y_{i,t}\right)=\sum_{t=1}^{n}D_{i,t},\;\forall i$$
(22)$$\sum_{t^{\prime }=1}^{t}\left(X_{i,t^{\prime }}\cdot y_{i,t^{\prime }}\right)\geq \sum_{t^{\prime }=1}^{t}D_{i,t^{\prime }},\;\forall i,t$$


**Proof.** Since we have the assumption that both the initial inventory and the final inventory are 0, Equation (21) must be correct because it represents a balance of the total output of all the planned periods and the total demand that guarantees the optimality of the solution. Equation (13) guarantees that at any period, the total demand of customers should be satisfied by the production volume, which is always satisfied by any feasible solution, including the optimal solution. □

## 4. Computational Experiments

The computational experiments were conducted on an Apple MacBook Pro (Apple Inc, Cupertino, CA, USA) with an Intel Core i7 of 2.9 GHz and a RAM size of 8 GB. The operating system was OS X 10.8.5 (Apple Inc, Cupertino, CA, USA). The models were coded by the AMPL software and solved by IBM CPLEX (version 12.6.0.1, Lucent Technologies, Murray Hill, NJ, USA).

### 4.1. Optimality Test with Small-Sized Problems

In order to verify the solution effect of the MILP model for the CMLLS-TVE problem, we used AMPL/CPLEX to solve the 96 well-known small-sized test problem instances from Dellaert and Jeunet (2000), which contains five items with an assembly structure over a 12-period planning horizon. Since the original benchmark instances do not have capacity constraints, we gave capacity constraints (*M_it_, α_i_, u_it_*) for the 96 instances to make them adaptable to the CMLLS-TVE model. We repeatedly ran the AMPL/CPLEX solver five times for each problem.

The solution results of the 96 small-sized TV-MLCLSP problem instances with five products over 12 planned periods are shown in Table 2. Through these computational experiments, the proposed MILP model was verified for its effectiveness and efficiency on solving the TV-MLCLSP. For all 96 examples, the optimal solution can always be obtained in a few seconds of CPU time. The average running time was 5.8 s, and the longest computation time was only 18.4 s. This shows that the proposed MILP model can solve efficiently the CMLLS-TVE problem with optimality for small-sized instances.

### 4.2. Efficiency Test with Medium-Sized Problems

In order to verify whether the proposed models can effectively solve the CMLLS-TVE problem of a medium-size, we tested them with 40 medium-sized problem instances from Dellaert and Jeunet [12,13], which were generated based on the product structures in Afentakis et al. [1] and Afentakis and Gavish [2]. Each of the medium-sized problem had 40–50 products (and parts) with general assembly structures over a 12/24-period planning horizon. Based on the original problem instances, we tested the sub-problem instances with combinations of 5, 10, 20, and 40 products and 6, 12, 18, and 24 planning periods. The CPU times used are listed in Table 3 for the average of five repeated runs. A general trend was recognized, where the average CPU time increased sharply as the number of products or the number of planning periods increased. When the problem size was larger than 20 products and 18 planning periods, the optimal solution could no longer be delivered by the MIP solver within a 2-h time limit.

As seen in Table 3, with the increase in the product number and planning periods, the solution time increased dramatically. This was because the CMLLS-TVE problem is NP-hard and the CPLEX solver is based on the branch-and-bound algorithm for solving an MIP problem. Variables in the CMLLS-TVE model, including *y_it_, D_it_, I_it_, X_it_,* and *u_it_*, all have a m×n solution space. Thus, the increase in the product number and planning periods led to an exponential growth in the solution time. Therefore, for large-sized CMLLS-TVE problems, only heuristic algorithms can be used to search for good-enough feasible solutions, instead of the optimal ones.

### 4.3. A Fix-and-Optimize Heuristic Approach for Large-Sized Problems

Fix-and-optimize heuristics is a concrete realization of a partial optimization strategy. Its basic principle is described as follows. For large-sized complex problems with multiple decision variables that cannot be solved with optimality in an acceptable time, partial optimization can be applied to optimize only a small part of selected decision variables while fixing most of the others with given values. Thus, the selected decision variables can be efficiently optimized in very short CPU time, and then repeated to select another part of the decision variables to fulfill similar partial optimizations iteratively, until no further improvements can be made after a given number of continuous attempts.

Hence, for large-sized CMLLS-TVE problems, we developed a fix-and-optimize heuristic to find a good-enough feasible solution within a controllable CPU time. In the CMLLS-TVE, variable *y_it_* is an independent decision variable on which all of the rest of the variables, e.g., *X_it_*, *I_it_*, *D_it_*, ect., are dependent. Therefore, we choose *y_it_* as the target variable to be executed with the partial optimization strategy. In such a fix-and-optimize algorithm, we first initialize a feasible solution as the incumbent solution, and then iteratively improve the incumbent solution by performing selecting, unfixing, and optimizing on variable *y_it_*. Before we provide the detailed steps of the algorithm, the following definition is given.

**Definition** **4.**
*Neighborhoods C_k_(Ω) or R_k_(Ω) of an incumbent solution: A neighborhood of an incumbent solution is defined by C_k_(Ω) or R_k_(Ω), indicating the set of neighboring solutions that may have different values in decision matrix y_it_ specified by k columns at Ω direction, where Ω indicates either a horizontal or vertical direction.*


In CMLLS-TVE, the variable *y_it_* is a m×n dimension matrix. Parameter *k* represents the number of rows (or columns) selected to be unfixed at one time. In order to guarantee the accuracy and efficiency of the solution, we let *k* = 1, 2, and 3, and let *Ω* indicate the row (or column) with value by *ω* (or *ω′*). Notion *ω* represents a sequential selection and *ω′* represents a random selection. We define set Items={1,2,…,m} and set Periods={1,2,…,n}. Then A⊆Items,|A|=k, and set B⊆Periods,|B|=k. Mathematical expressions of *C_k_(**Ω)* and *R_k_(**Ω)* are shown as follows.

(23)$$C_{k}(\Omega )=\left\{y_{it}|i\in A,A\subseteq Items,\left|A\right|=k\right\}$$

(24)$$R_{k}(\Omega )=\left\{y_{it}|t\in B,B\subseteq Periods,\left|B\right|=k\right\}$$

Thus, we can combine four operators of neighborhood selection in Table 4, as shown.

Based on the above four operators, given the parameter *K*_max_ of the maximum number of columns (or rows) and the parameter *N*_max_ of the maximum continuous attempts, we can implement the fix-and-optimize algorithm on the CMLLS-TVE, described as Algorithm 1.

**Algorithm 1.** Scheme of fix-and-optimize algorithm.
(1)Generate an initial solution by assigning $y_{it}\leftarrow 1,\;\forall i,t$ as the incumbent solution.(2)Let *k*←1, *N*←0.(3)Repeat the following Steps (a), (b), (c), and (d), until *k* = *K*_max_ or the time limit is met:(a)Select randomly one of the four operators $C_{k}(\omega )$, $C_{k}(\omega^{\prime })$, $R_{k}(\omega )$, $R_{k}(\omega^{\prime })$ to unfix a partial of variable instances *y_it_* and fix the rest of the others.(b)Call a MIP solver to implement partial optimization.(c)If the new solution is better than the incumbent solution, then update the incumbent solution and let *N*←0; otherwise, let *N*←*N* + 1.(d)If N >= *N*_max_, then let *k*←*k* + 1 and *N*←0.(4)Output the incumbent solution and stop.


### 4.4. Computational Experiment with the Fix-and-Optimize Heuristics

The benchmark problem instances data from Dellaert and Jeunet (2000) were used to test the proposed fix-and-optimized heuristic algorithm. The problem set contains 40 instances that have 40–50 products over a planning horizon of 12/24 periods. However, since neither the productivity restricts nor the time-varying setup cost were considered in these instances, we gave a large value to the *M_it_, α_i_, u_it_*, and a zero-increasing rate of the setup cost, in order to transform the original MLLS problem instances into CMLLS-TVE instances. To verify the effectiveness of the proposed fix-and-optimize heuristic approach, we set the time limit to be 60 s and compared them with the results obtained by CPLEX with a time limit of 1800 s (by adding a statement in AMPL: option CPLEX options “timelimit = 1800”). 

In Table 5, we show the results of the computational experiments and the comparisons. The product structures numbers 1, 2, and 3 represent the assembly structure, the sequential structure, and the general structure of the problem instances, respectively. The number of independent end-products was three. For product structure number 4, the number of independent end-products were four. Column Δ*TC* indicates the differences between the results of the fix-and-optimize algorithm in 60 s and the results of CPLEX in 1800 s. If the number ΔTC is negative, then the solution of the fix-and-optimize algorithm was better than that of CPLEX. It can be observed that for 32 out of 40 instances, the fix-and-optimize algorithm had a better solution by 60 s than that of CPLEX. The average deviation was −733. Therefore, for medium and large-sized CMLLS-TVE instances, the proposed fix-and-optimize algorithm is applicable to get good-enough solutions within a controllable computational time. We use column Δ*R* to indicate the amplitude of entropy reduction after optimization. Because the complexity of product structure and the demand of product quantity remain unchanged before and after optimization, Δ*R* depends only on the change of objective cost. Therefore, the calculation formula for the system entropy can be described as:(25)$$\Delta R=(OC_{B}-OC_{A})/OC_{A}$$

The above Equation (25) provides a method for calculating entropy change rate of the production system, where symbol *OC_A_* represents the production cost of optimized production system and symbol *OC_B_* represents the production cost of original production system.

## 5. Conclusions

This study investigated the capacitated multi-level lot-sizing problem under a time-varying environment (CMLLS-TVE) and extended the traditional MLLS model by considering time-varying production factors. We considered time-varying production setup cost and time-varying production capacities and proposed a non-linear mathematical programming model and a mixed-integer linear programming model for the CMLLS-TVE. In the proposed models, the time-varying setup cost and dynamic capacity constraint were taken as new constraints. New properties about the time-varying characteristics in CMLLS-TVE were analyzed. In order to solve the medium- and large-sized problems with computational efficiency and good solution quality, we developed a fix-and-optimize solution approach implemented in the AMPL/CPLEX environment. Computational experiments on well-known benchmark problem instances in the literature showed that the proposed algorithms can effectively solve the small-sized CMLLS-TVE problems with optimal solutions, while the fix-and-optimal heuristic was able to find near-optimal solutions with controllable computational time for medium- and large-sized CMLLS-TVE problems. After optimization, the system entropy showed a decreased value, indicating that the optimization made the production in the system become more orderly. Future studies can be carried out on four aspects: (1) to make the CMLLS-TVE more practical by considering more time-varying factors, (2) to develop parallel heuristic algorithms with higher efficiency for very large-sized CMLLS-TVE problem instances, (3) to develop a more reasonable and accurate method for calculating system entropy, and (4) to combine the economic lot-sizing problem with the optimizations of plant layout, process flow, and logistics and model the optimization in view of the whole production system.

## Figures and Tables

**Table 1 entropy-21-00377-t001:** Symbolic definitions of the capacitated MLLS problem under a time-varying environment (CMLLS-TVE) model.

Symbol	Explanation
Parameter:	
*m*	Total number of products (including parts)
*n*	Total number of production periods over the planning horizon
*i*	Index of product (including parts)
*t*	Index of production period
*s_i,t_*	Setup cost of product *i* during the period *t*
*H_i,t_*	Unit cost of inventory-holding for product *i* in period *t*
*M_i,t_*	Maximum capacity of product *i* that can be produced in period *t*
*d_i,t_*	External requirements (from customers) of product *i* during the period *t*
*D_i,t_*	Total demand of product *i* in period *t,* including external demand and internal demand
*C_i,j_*	The product structure, indicating the number of products *i* used to produce product *j*
*Γ_i_*	Set of direct successor products of product *i*
$\Gamma_{i}^{-1}$	Set of direct predecessor products of product *i*
*l_i_*	lead time of product *i*, if needed in period *t*, then production must be arranged in period *t* − *l_i_*.
*α_i_*	Growth coefficient of setup cost for product *i*
*Decision variables:*	
*y_it_*	Binary decision variable, indicating if product *i* is scheduled for production in period *t*
*I_it_*	Non-negative decision variable, indicating the inventory of product *i* at the end of period *t*
*X_it_*	Number of products *i* arranged to produce, started in the period *t* and finished in period *t* + *l_i_*
*L_it_*	Production interval of product *i* in period *t*

**Table 2 entropy-21-00377-t002:** Calculation results of the 96 small-scale problem instances (CMLLS-TVE).

Problem No.	Optimal Solution	Problem No.	Optimal Solution	Problem No.	Optimal Solution	Problem No.	Optimal Solution
0	984.93	24	1245.64	48	1482.96	72	1402.48
1	846.38	25	1002.91	49	1380.09	73	1515.90
2	1059.78	26	875.80	50	1207.79	74	1365.47
3	855.29	27	1405.56	51	1631.69	75	1689.09
4	786.75	28	1275.42	52	1239.53	76	1646.16
5	902.73	29	829.89	53	978.94	77	1576.88
6	1247.28	30	1328.52	54	1733.47	78	1830.30
7	1279.51	31	1173.84	55	1474.13	79	1828.76
8	1018.47	32	1403.66	56	1767.20	80	1739.96
9	1154.90	33	1560.80	57	1826.97	81	1666.15
10	937.83	34	1559.91	58	1420.11	82	1440.70
11	971.84	35	1032.96	59	1591.52	83	1472.72
12	1167.12	36	1523.27	60	1426.32	84	2022.54
13	1074.67	37	1518.01	61	1616.45	85	1921.94
14	979.37	38	1210.94	62	1737.39	86	1594.65
15	1183.37	39	1396.97	63	1704.36	87	1637.68
16	787.13	40	1067.77	64	1541.81	88	1484.71
17	838.00	41	1049.42	65	1568.04	89	1657.46
18	1819.32	42	2248.63	66	2154.69	90	1942.97
19	2052.23	43	2334.54	67	1808.15	91	1959.45
20	1825.04	44	1695.17	68	1965.99	92	1728.54
21	1718.71	45	2028.77	69	1760.67	93	2190.22
22	1711.30	46	2044.11	70	1600.15	94	2054.16
23	1699.48	47	1817.31	71	1853.75	95	1579.07

**Table 3 entropy-21-00377-t003:** Average CPU time used with respect to different product numbers and planned periods.

Number of Periods	Number of Products			
	5	10	20	40
6	0.34 s	4.7 s	53 s	27 min
12	5.8 s	45 s	8.2 min	30 min
18	53 s	43 min	2.3 h	>5 h
24	1.4 min	1.8 h	>5 h	>>5 h

**Table 4 entropy-21-00377-t004:** Explanation of the removing fixation scheme.

Symbol	Explanation of Scheme
$C_{k}(\omega )$	Select *k* columns in order
$C_{k}(\omega^{\prime })$	Select *k* columns randomly
$R_{k}(\omega )$	Select *k* rows in order
$R_{k}(\omega^{\prime })$	Select *k* rows randomly

**Table 5 entropy-21-00377-t005:** Results of computational experiment on 40 instances of CMLLS-TVE.

ID.	Description			Fix-and-Optimize Heuristics		CPLEX		ΔTC.	ΔR
	Structure	Product	Period	Obj. Cost		Obj. Cost	CPU Time		
0	1	50	12	**195764**	−1476	197240	1800	−1476	−1.02%
1	1	50	12	167156	537	**166619**	1800	537	−1.80%
2	1	50	12	**201519**	−1345	202864	1800	−1345	−1.98%
3	1	50	12	**189944**	−741	190685	1800	−741	−1.32%
4	1	50	12	**162857**	60.0	163238	1800	−381	−2.15%
5	2	50	12	**181233**	60.0	182551	1800	−1318	−2.48%
6	2	50	12	**157213**	60.0	159341	1800	−2128	−1.27%
7	2	50	12	**184309**	60.0	185547	1800	−1238	−1.63%
8	2	50	12	**137762**	60.0	138492	1800	−730	−2.90%
9	2	50	12	**188358**	60.0	189677	1800	−1319	−1.33%
10	3	40	12	**150014**	60.0	151488	1800	−1474	−2.33%
11	3	40	12	199725	60.0	**199584**	153.8	141	−2.25%
12	3	40	12	**162963**	60.0	163877	1800	−914	−2.34%
13	3	40	12	**186729**	60.0	187676	1800	−947	−2.09%
14	3	40	12	**162666**	60.0	163859	1800	−1193	−2.45%
15	4	40	12	**186415**	60.0	187788	1800	−1373	−2.19%
16	4	40	12	**186770**	60.0	187118	1800	−348	−2.23%
17	4	40	12	**194375**	60.0	195441	1800	−1066	−2.19%
18	4	40	12	**137810**	60.0	139812	1800	−2002	−3.15%
19	4	40	12	168149	60.0	**166865**	1247.6	1284	−2.65%
20	1	50	24	345240	60.0	**344573**	1800	667	−1.31%
21	1	50	24	**294497**	60.0	294740	1800	−243	−1.56%
22	1	50	24	**356096**	60.0	357664	1800	−1568	−1.32%
23	1	50	24	**326286**	60.0	327804	1800	−1518	−1.46%
24	1	50	24	388415	60.0	**387878**	1800	537	−1.25%
25	2	50	24	**342872**	60.0	344100	1800	−1228	−1.45%
26	2	50	24	380879	60.0	**380733**	1800	146	−1.32%
27	2	50	24	**348082**	60.0	349071	1800	−989	−1.47%
28	2	50	24	**414460**	60.0	415030	1800	−570	−1.26%
29	2	50	24	**391856**	60.0	392551	1800	−695	−1.35%
30	3	40	24	**347138**	60.0	347182	1800	−44	−1.55%
31	3	40	24	**353882**	60.0	354731	1800	−849	−1.55%
32	3	40	24	**357991**	60.0	359584	1800	−1593	−1.55%
33	3	40	24	413653	60.0	**413482**	53.7	171	−1.37%
34	3	40	24	404030	60.0	**403717**	85.4	313	−1.42%
35	4	40	24	**291120**	60.0	291729	1800	−609	−2.00%
36	4	40	24	**339370**	60.0	339774	1800	−404	−1.74%
37	4	40	24	**322227**	60.0	322871	1800	−644	−1.86%
38	4	40	24	**368075**	60.0	369409	1800	−1334	−1.65%
39	4	40	24	**306287**	60.0	307136	1800	−849	−2.02%
Average value				264855	60.0	265588	1658.5	−733	−1.81%

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
