# Peer review of "Modeling the Capacitated Multi-Level Lot-Sizing Problem under Time-Varying Environments and a Fix-and-Optimize Solution Approach"

_entropy, 2019, doi:10.3390/e21040377_

Round 1
Reviewer 1 Report
The paper is of interesting and important topic. The problem of optimization of production flow is complex because both of the dynamic character of the production process and NP-hard character of the optimization task. Therefore the attempt to provide a new solution is undoubtedly valuable.
The Authors investigate the capacited multilevel lot-sizing problem under a time-varying environment and present a new mixed-integer linear programming model for the analyzed problem. The provided mathematical model seems to be correct. The realized computational experiments prove its efficiency. Unfortunately the solution seems to be more theoretical than practical. Moreover, in my opinion the Authors should emphasize the role of the presented model in the manufacturing processes organization. In fact I have some suggestions that can/should be taken into account before publication.
1) When you describe the entropy in production systems you only mention that the “process of optimizing production schedule is to make the production system more orderly”. In fact it is very important to prepare optimized and robust production plans and schedules to reduce the necessity of re-scheduling during the manufacturing process and it should be more emphasized. Therefore I suggest to describe shortly the problem of robust planning and scheduling and give some references to publications that describe this problem (see: eg.: Tolio T., Urgo M., Vancza J., Robust production control against propagation of disruptions, CIRP Annals, Vol. 60, Issue 1, 2011, pp. 489-492; Sobaszek Ł., Gola A., Świć A., Predictive scheduling as a part of intelligent job scheduling system, Advances in Intelligent Systems and Computing, Vol. 637, 2018, pp. 358-367.
2) When you talk about the optimization in the context of multilevel lot-sizing problem you should remember that the problem cannot be treated only as a production planning problem but that is also connected with the other subsystems of manufacturing system such as internal transport, handling etc. Therefore I suggest to give short description of this problem and give some literature examples where similar problems were analyzed (see. eg.: Gola A., Kłosowski G., Application of fuzzy logic and genetic algorithms in automated works transport organization, Advances in Intelligent Systems and Computing, Vol. 620, 2018; Bocewicz G.K., Nielsen I.E., Smutnicki C., Banaszak Z., Towards the leveling of multi-product bath production flow. A multimodal networks perspective, IFAC PapersOnLine 51-11, 2018, pp. 1434-1441).
3) When you make a literature review you describe a lot of different methods that are used for optimization problems in the production environment. Please note that because of the character of the problem despite of mathematical methods the simulation methods are frequently used (see eg.: Kłos S., Patalas-Maliszewska J., Trebuna P., Improving manufacturing processes using simulation methods, Applied Computer Science, Vol. 12, No. 4, 2016, pp. 7-17; Kikolski M., Idenfication of production bottlenecks with the use of plant simulation software, Economics and Management, Vol. 8, Issue 4, 2016, pp. 103-112; Gola A., Reliability analysis of reconfigurable manufacturing system structures using computer simulation methods, Eksploatacja I Niezawodnosc – Maintenance and Reliability 21(1), 2019, pp. 90-102). In fact – to prove the real value of your model it would be valuable to evaluate obtained results making some simulation experiments. I understand that because of the limited range of the paper it can be difficult to do it now. But you should at least mention about this methods in your paper. And maybe it is good idea to make such experiments for further paper?
To summarize – In my opinion the scientific value of the paper is good and the presented model can be interesting for other researchers. Therefore it can be published. Unfortunately the paper focuses too much on the problem without “general description” what is its “place” in real manufacturing systems what makes it too theoretical. Therefore some additional descriptions (mentioned in above remarks) are suggested.
Author Response
We thank you very much for your valuable comments and suggestions, which have helped to improve the quality and presentation of the paper substantially. Followings are point-wise responses to your comments and suggestions. All modifications to the manuscript had also been highlighted in the revised submission.
The paper is of interesting and important topic. The problem of optimization of production flow is complex because both of the dynamic character of the production process and NP-hard character of the optimization task. Therefore the attempt to provide a new solution is undoubtedly valuable.
The Authors investigate the capacited multilevel lot-sizing problem under a time-varying environment and present a new mixed-integer linear programming model for the analyzed problem. The provided mathematical model seems to be correct. The realized computational experiments prove its efficiency. Unfortunately the solution seems to be more theoretical than practical. Moreover, in my opinion the Authors should emphasize the role of the presented model in the manufacturing processes organization. In fact I have some suggestions that can/should be taken into account before publication.
1) When you describe the entropy in production systems you only mention that the “process of optimizing production schedule is to make the production system more orderly”. In fact it is very important to prepare optimized and robust production plans and schedules to reduce the necessity of re-scheduling during the manufacturing process and it should be more emphasized. Therefore I suggest to describe shortly the problem of robust planning and scheduling and give some references to publications that describe this problem (see: eg.: Tolio T., Urgo M., Vancza J., Robust production control against propagation of disruptions, CIRP Annals, Vol. 60, Issue 1, 2011, pp. 489-492; Sobaszek Ł., Gola A., Świć A., Predictive scheduling as a part of intelligent job scheduling system, Advances in Intelligent Systems and Computing, Vol. 637, 2018, pp. 358-367.
Responses:Thank you for this comment. Yes, it is true that traditional lot-sizing scheduling algorithms did not consider the time-varying production factors that often lead to re-scheduling and re-optimizing of the production plans. This is also our initial motivation to carry out this study. We had added some relevant content in the revision paper, and cited the papers you recommended Thank you. The added section is highlighted in the page 4 ((lines 161~163) and 15 (lines 554, 560).
2) When you talk about the optimization in the context of multilevel lot-sizing problem you should remember that the problem cannot be treated only as a production planning problem but that is also connected with the other subsystems of manufacturing system such as internal transport, handling etc. Therefore I suggest to give short description of this problem and give some literature examples where similar problems were analyzed (see. eg.: Gola A., Kłosowski G., Application of fuzzy logic and genetic algorithms in automated works transport organization, Advances in Intelligent Systems and Computing, Vol. 620, 2018; Bocewicz G.K., Nielsen I.E., Smutnicki C., Banaszak Z., Towards the leveling of multi-product bath production flow. A multimodal networks perspective, IFAC PapersOnLine 51-11, 2018, pp. 1434-1441).
Responses:Thank you for this suggestion. Yes, the multilevel lot-sizing (MLLS) problem plays an important hub-role in the production system of a manufacturing firm. Actually, for each component of the production structure, we associate it with a leading time which is needed by subsystems like internal transports, material handling, purchases, etc. Thus, the whole manufacturing system are integrated to be optimized in the MLLS model. We have added relevant content and cited the works you recommended in the revision version of our paper. The added section is highlighted in the pages 3(lines 110~113), 5 (lines 241-244), 13( line 491) and 14( line 513).
3) When you make a literature review you describe a lot of different methods that are used for optimization problems in the production environment. Please note that because of the character of the problem despite of mathematical methods the simulation methods are frequently used (see eg.: Kłos S., Patalas-Maliszewska J., Trebuna P., Improving manufacturing processes using simulation methods, Applied Computer Science, Vol. 12, No. 4, 2016, pp. 7-17; Kikolski M., Idenfication of production bottlenecks with the use of plant simulation software, Economics and Management, Vol. 8, Issue 4, 2016, pp. 103-112; Gola A., Reliability analysis of reconfigurable manufacturing system structures using computer simulation methods, Eksploatacja I Niezawodnosc – Maintenance and Reliability 21(1), 2019, pp. 90-102). In fact – to prove the real value of your model it would be valuable to evaluate obtained results making some simulation experiments. I understand that because of the limited range of the paper it can be difficult to do it now. But you should at least mention about this methods in your paper. And maybe it is good idea to make such experiments for further paper?
Responses:Thank you for this comment. We have added relevant content and cited the literature you recommended. The added section is highlighted in the 3 (lines 113~115) and 14 (lines 515, 525, and 527).
To summarize – In my opinion the scientific value of the paper is good and the presented model can be interesting for other researchers. Therefore it can be published. Unfortunately the paper focuses too much on the problem without “general description” what is its “place” in real manufacturing systems what makes it too theoretical. Therefore some additional descriptions (mentioned in above remarks) are suggested.

Reviewer 2 Report
The article deals with a multi-level problem that is important for modern production and inventory theory. The problem is formulated in the form of optimization, where the NP
hardness is an almost insurmountable complication. In other words, it means that only the smallest variants of systems can be solved/optimized with the exact (combination)
optimization schemes that are available at the moment.
In line with modern trends in the analysis of biological and complex systems, the authors and also many researchers believe that some emerging problems can be appropriately formulated as aspects of variable environments that are typical of many evolving and real systems in general. Thus, the concept appears in many biology and optimization works, it also has an influence on thinking in the economic, industrial and financial fields. (I expect readers to be interested in this topic.)
There are many optimization tasks that people have recently been trying to transform and solve for variable environments. Corresponding procedures are generally very interesting but also quite diverse, since the formulation for variable environments itself opens up many opportunities for alternative modeling, implementation and interpretation of time variability. I agree with the authors who argue that the main time-varying factors may be changes in the market with costs / prices, customers requirements, availability of production resources and capacities.
Conditions of temporal variability can also have important consequences of the clarity and understanding of the results provided. Agree with the authors that they are trying to evaluate their optimization results in terms of entropy.
Section 2.1 discusses the introduction of entropy. But it gave me a strange feeling. It is not a systematic description of the problem at all and it is not very clear. There is obviously no explicit formula for calculating entropy.
According to the authors, this is not the sole approach to entropy. Other perspectives exist. I would also very much like to see different conceptual alternatives, particularly with regard to the nature of the Entropy journal where this would be an important question on the overall focus of this work. (Note that the decrease in entropy during optimization is something that is intuitively clear and consistent with the thermodynamic / information expectations for the target - objective function value.)
It should be emphasized that many types of variable environments are generally considered in modeling and biological literature. The basic classification distinguishes, for example, between periodic, non-periodic, deterministic, stochastic or predictable environments. In some more complex cases, exogenous variability may be affected by collective feedback from previous (delayed) dynamic activities. I think that the submitted model classification is not detailed enough with regard to the above general terms. For example, it is unclear whether the study deals with the exogenous or endogenous environmental dynamics. These aspects also need to be thoroughly discussed in terms of the model's functions and purposes.
Some technical note: Line 291. There is I think the absence of indices in y variable.
Author Response
Response to Reviewer 2 Comments
We thank you very much for your valuable comments and suggestions, which have helped to improve the quality and presentation of the paper substantially. Followings are point-wise responses to your comments and suggestions. All modifications to the manuscript had also been highlighted in the revised submission.
The article deals with a multi-level problem that is important for modern production and inventory theory. The problem is formulated in the form of optimization, where the NP
hardness is an almost insurmountable complication. In other words, it means that only the smallest variants of systems can be solved/optimized with the exact (combination)
optimization schemes that are available at the moment.
In line with modern trends in the analysis of biological and complex systems, the authors and also many researchers believe that some emerging problems can be appropriately formulated as aspects of variable environments that are typical of many evolving and real systems in general. Thus, the concept appears in many biology and optimization works, it also has an influence on thinking in the economic, industrial and financial fields. (I expect readers to be interested in this topic.)
There are many optimization tasks that people have recently been trying to transform and solve for variable environments. Corresponding procedures are generally very interesting but also quite diverse, since the formulation for variable environments itself opens up many opportunities for alternative modeling, implementation and interpretation of time variability. I agree with the authors who argue that the main time-varying factors may be changes in the market with costs / prices, customers requirements, availability of production resources and capacities.
Conditions of temporal variability can also have important consequences of the clarity and understanding of the results provided. Agree with the authors that they are trying to evaluate their optimization results in terms of entropy.
Point 1: Section 2.1 discusses the introduction of entropy. But it gave me a strange feeling. It is not a systematic description of the problem at all and it is not very clear. There is obviously no explicit formula for calculating entropy.
According to the authors, this is not the sole approach to entropy. Other perspectives exist. I would also very much like to see different conceptual alternatives, particularly with regard to the nature of the Entropy journal where this would be an important question on the overall focus of this work. (Note that the decrease in entropy during optimization is something that is intuitively clear and consistent with the thermodynamic / information expectations for the target - objective function value.)
Response 1: Thank you for this comment. We revised some of the contents in Section 2.1 according to your suggestions. We referred to the basic formulas of entropy change and gave the formulas of entropy change rate that is accord with the objective of this study. In addition, we revised the last part of Section 4.4 and simplified the calculation formula of entropy variability according to the actual situation. The modified part is highlighted in page 5(204~215) and page 11(440~448).
Point 2: It should be emphasized that many types of variable environments are generally considered in modeling and biological literature. The basic classification distinguishes, for example, between periodic, non-periodic, deterministic, stochastic or predictable environments. In some more complex cases, exogenous variability may be affected by collective feedback from previous (delayed) dynamic activities. I think that the submitted model classification is not detailed enough with regard to the above general terms. For example, it is unclear whether the study deals with the exogenous or endogenous environmental dynamics. These aspects also need to be thoroughly discussed in terms of the model's functions and purposes.
Response 2: Thank you for this correction. We revised the introduction section to reflect the discussion of variable environments. At the same time, what we study is exogenous environmental dynamics. The modified part is highlighted in page 3(119~123) and page 4(189~193).
Point 3: Some technical note: Line 291. There is I think the absence of indices in y variable.
Response 3: Thank you for this correction. We corrected the expression in Property 2 where an indic i is missed in variable y. Thank you.

Reviewer 3 Report
Interesting research, but ...
1. Entropy definition added without any justification with the aim of the work.
"This is a preliminary idea, which needs further improvement" I agree.
2. The real purpose of the article is to present the results of the calculation of the modified lot-sizing problem. In other words, the article presents the results of the computational experiment of the formulated task and the proper conclusion is that it can be solved.
3. The innovation: time-varying setup cost in the model. Is the modification is justified by the actual requirements of the real system, in which industries, in which specific production process it is reasonable to introduce it?
4. References to fairly old publications.
5. English language and style needs to be corrected.
6. Future research - too trivial.
Author Response
Response to Reviewer 3 Comments
We thank you very much for your valuable comments and suggestions, which have helped to improve the quality and presentation of the paper substantially. Followings are point-wise responses to your comments and suggestions. All modifications to the manuscript had also been highlighted in the revised submission.
Interesting research, but ...
Point 1: Entropy definition added without any justification with the aim of the work.
"This is a preliminary idea, which needs further improvement" I agree.
Response 1: Thank you for this comment. In this paper, entropy change and entropy change rate are intuitive criteria for evaluating the effectiveness of optimization. Based on your suggestion, we have revised the content of the article about entropy. The modified part is highlighted in page 5 (lines 202~215), page 11 (lines 440~448) and page 13 (lines 463~469).
Point 2: The real purpose of the article is to present the results of the calculation of the modified lot-sizing problem. In other words, the article presents the results of the computational experiment of the formulated task and the proper conclusion is that it can be solved.
Response 2: Thank you for this correction. Our motivation is to build a linear mathematical model for the multilevel lot-sizing problem considering time-varying environment and capacity constraints, and to develop algorithms for solving small-scale and large-scale problems. Then, we use the theory of system entropy to evaluate our optimization effect.
Point 3: The innovation: time-varying setup cost in the model. Is the modification is justified by the actual requirements of the real system, in which industries, in which specific production process it is reasonable to introduce it?
Response 3: Thank you for this comment. Typical examples of production process that has time-varying setup cost had been given in lines 189-191.
Point 4: References to fairly old publications.
Response 4: Thank you for this comment. We updated the references with seven new references, six of them published after 2015 and one published in 2011, which are highlighted in the reference section.
Point 5: English language and style needs to be corrected.
Response 5: Thank you for this comment. We have read through the paper one more times, corrected some typo errors and improved multiple places of the gramor expressions.
Point 6: Future research - too trivial.
Response 6: Thank you for this comment. We add one more future research direction: (4) to combine the economic lot-sizing problem with the optimization problem of plant layout, process flow and logistics and model the optimization of the whole production system. This is highlighted in page 12(450~451).

Round 2
Reviewer 2 Report
In general, I agree with the changes done by the authors. But I still have one little comment they could work on.
The authors have elaborated a literature review on the entropy problem, and I think it is very well done on lines manuscript 157-169. Nevertheless, it is not entirely clear how knowledge of the literature cited is later reflected in the entropy form presented in section 2.1, where references are no longer mentioned. Authors should try to identify more precisely the works that influenced them in the choice of the term Delta R term.
Author Response
We thank you very much for your valuable comments and suggestions, which have helped to improve the quality and presentation of the paper substantially. Followings are point-wise responses to your comments and suggestions. All modifications to the manuscript had also been highlighted in the revised submission.
In general, I agree with the changes done by the authors. But I still have one little comment they could work on.
Point 1: The authors have elaborated a literature review on the entropy problem, and I think it is very well done on lines manuscript 157-169. Nevertheless, it is not entirely clear how knowledge of the literature cited is later reflected in the entropy form presented in section 2.1, where references are no longer mentioned. Authors should try to identify more precisely the works that influenced them in the choice of the term Delta R term.
Response 1: Thank you for this comment. All the cited articles have played a certain role in the part of the paper about entropy. Generally speaking, Martina et.al, 2019 and Qiuxiang et.al, 2018 have a greater impact on Section 2.1. I have mentioned these two papers in section 2.1 and highlight the modified part (WORD lines 204~205).

Reviewer 3 Report
It looks better after corrections, but you can always improve it. However, in this form it is acceptable.
Author Response
We thank you very much for your valuable comments and suggestions, which have helped to improve the quality and presentation of the paper substantially. Followings are point-wise responses to your comments and suggestions. All modifications to the manuscript had also been highlighted in the revised submission.
Point1: English language and style are fine/minor spell check required
Response 1: Thank you for this comment. We read the article and revised the spelling, format and grammar again.
Point 2: It looks better after corrections, but you can always improve it. However, in this form it is acceptable.
Response 2: Thank you for this comment. We mentioned two papers (Martina et.al, 2019 and Qiuxiang et.al, 2018) in section 2.1, because of their contribution to this section.
